# Cytoprotective Role of Heme Oxygenase-1 in Cancer Chemoresistance: Focus on Antioxidant, Antiapoptotic, and Pro-Autophagy Properties

**DOI:** 10.3390/antiox12061217

**Published:** 2023-06-05

**Authors:** Huan Wang, Qi Cheng, Lingjie Bao, Mingqing Li, Kaikai Chang, Xiaofang Yi

**Affiliations:** 1Department of Gynecology, Hospital of Obstetrics and Gynecology, Fudan University, Shanghai 200011, China; 2Shanghai Key Laboratory of Female Reproductive Endocrine Related Diseases, Shanghai 200011, China

**Keywords:** heme oxygenase-1 (HO-1), reactive oxygen species, cancer, chemoresistance, cytoprotective effect, antioxidant, apoptosis, autophagy

## Abstract

Chemoresistance remains the foremost challenge in cancer therapy. Targeting reactive oxygen species (ROS) manipulation is a promising strategy in cancer treatment since tumor cells present high levels of intracellular ROS, which makes them more vulnerable to further ROS elevation than normal cells. Nevertheless, dynamic redox evolution and adaptation of tumor cells are capable of counteracting therapy-induced oxidative stress, which leads to chemoresistance. Hence, exploring the cytoprotective mechanisms of tumor cells is urgently needed to overcome chemoresistance. Heme oxygenase-1 (HO-1), a rate-limiting enzyme of heme degradation, acts as a crucial antioxidant defense and cytoprotective molecule in response to cellular stress. Recently, emerging evidence indicated that ROS detoxification and oxidative stress tolerance owing to the antioxidant function of HO-1 contribute to chemoresistance in various cancers. Enhanced HO-1 expression or enzymatic activity was revealed to promote apoptosis resistance and activate protective autophagy, which also involved in the development of chemoresistance. Moreover, inhibition of HO-1 in multiple cancers was identified to reversing chemoresistance or improving chemosensitivity. Here, we summarize the most recent advances regarding the antioxidant, antiapoptotic, and pro-autophagy properties of HO-1 in mediating chemoresistance, highlighting HO-1 as a novel target for overcoming chemoresistance and improving the prognosis of cancer patients.

## 1. Introduction

The development of resistance to chemotherapy is one of the main obstacles for tumor management. Although many in-depth studies have been conducted, the specific molecular mechanisms underlying this inevitable resistance remain to be fully elucidated. Currently, therapeutic strategies targeting reactive oxygen species (ROS) production to initiate cellular damage and programmed cell death in cancer treatment are an area of emerging interest [1,2,3]. ROS refers to highly reactive molecules produced by incomplete reduction of oxygen in normal metabolic processes [4], which are mainly categorized into two types, i.e., free radicals and nonradical ROS. Free radicals contain one or more unpaired electron(s) in their outer orbital, including superoxide radical anion (O_2_^●−^), hydroxyl radical (HO^●^), alkoxyl radical (RO^●^), and peroxyl radical (ROO^●^), while nonradical ROS, such as hydrogen peroxide (H_2_O_2_), organic peroxides (ROOH), and singlet oxygen (^1^O_2_), do not possess unpaired electrons but are the downstream products of incomplete reduction of molecular oxygen and have oxidizing potential [5,6]. The generation and elimination of ROS are balanced under physiological conditions, and excessive ROS can be scavenged by a tightly regulated endogenous antioxidant defense system consisting of enzymatic antioxidants such as superoxide dismutase (SOD), catalase (CAT), glutathione peroxidase (GPx), peroxiredoxins (Prx), thioredoxins(Trx), and nonenzymatic antioxidants including glutathione, uric acid, coenzyme Q, and lipoic acid [7,8]. ROS play an essential role in maintaining cell homeostasis by regulating intracellular signaling and the immune response at low concentrations. However, high levels of ROS are deleterious to nucleic acids, proteins, and cell membranes, and once an imbalance tends toward ROS overproduction, oxidative stress occurs, leading to cytotoxicity, cell dysfunction, and cell death [9,10]. Cancer cells are hypersensitive to fluctuations in ROS levels, which makes inducing ROS production over cytotoxic threshold levels an effective anticancer strategy [11].

Beyond causing cellular damage directly, ROS have also emerged as important signaling molecules in cancer cell fate decisions. The impact of ROS on the regulation of programmed cell death, such as apoptosis and autophagy, has been well established recently [12,13,14]. Remarkably, autophagy and apoptosis have been shown to interact with each other [15], while ROS may act as a molecular switch between autophagy and apoptosis [16]. Apoptosis, also defined as type I cell death, can be triggered by ROS via both the death receptor or extrinsic pathway and the mitochondrial or intrinsic pathway [17]. Several standard chemotherapeutic agents, including paclitaxel, cisplatin, bortezomib, and etoposide, have been revealed to achieve tumor therapeutic efficacy partially through initiating ROS-mediated apoptosis [18]. Adaptive autophagy is essential for cell survival, proliferation, differentiation, and homeostasis, while hyperactivation of autophagy results in type II cell death—autophagic cell death [19]. Interestingly, paradoxical effects have been described for ROS in autophagy. It was reported that excessive ROS levels can activate autophagic flux via the ROS-FOXO3-LC3/BNIP3, ROS-NRF2-P62, ROS-HIF1-BNIP3/NIX, and ROS-TIGAR pathways [20]. ROS accumulation was also shown to inhibit autophagy either by directly increasing AKT/mTOR or by indirectly transactivating antioxidant gene responses to oxidative stress, which serves as a feedback loop to repress autophagy [21]. Additionally, ROS are also relevant to other types of cell death, including ferroptosis [22], pyroptosis [23], and necroptosis [24]. ROS-based chemotherapy strategies have been widely adopted for clinical applications, yet exposure to chemotherapeutic drugs significantly stimulates the overactivation of the antioxidant defense system in cancer cells, which induces a high level of cellular redox balance, triggering adaptive responses to counteract the lethal effects of ROS [25]. Thus, the bright side and dark side of ROS rely on each other in cancer chemotherapy, and chemoresistance emerges when the cellular antioxidant capacity exceeds the cell killing capacity of ROS. In parallel, accumulating evidence suggests that antioxidative factor-targeted therapy can recover a low level of redox status in cancer cells and thus remodel the cytotoxic effect of chemotherapy agents [26,27]. Therefore, proteins engaged in antioxidant activities may play a crucial role in the occurrence of tumor chemoresistance.

Heme oxygenase 1 (HO-1), a rate-limiting enzyme that promotes the catalytic breakdown of heme into carbon monoxide (CO), free iron, and biliverdin, is an essential antioxidant enzyme that induces the host defense response to oxidative stress [28]. HO-1 is normally expressed at a low level under physiological conditions, while it can be activated in stressful situations, which is important for maintaining cellular homeostasis against exogenous stress and injury [29]. Nevertheless, HO-1 is a double-edged sword that plays a cytoprotective role not only in normal cells but also in tumor cells. HO-1 was indicated to be highly induced in multiple cancers, such as breast cancer [30], liver cancer [31], gastric cancer [32], colorectal cancer [33], esophageal squamous cell carcinoma [34], prostate cancer [35], pancreatic cancer [36], and neuroblastoma [37], and was closely related to poor patient prognosis. Upregulated expression and improved enzymatic activity of HO-1 not only represents a potent risk factor for tumorigenesis and tumor progression, but it also relates to therapy resistance [38]. HO-1 inhibition significantly improves chemosensitivity, which motivated an increasing number of preclinical studies committed to exploring and optimizing the ideal strategy for HO-1 targeting to resolve chemoresistance [39,40].

As a stress-related biosensor, the mechanisms of action of HO-1 in chemotherapy resistance are complicated. In addition to its antioxidant function, HO-1 has been revealed to be associated with apoptosis avoidance and autophagy induction, which both result in resistance to cell death [41,42,43]. Thus, approaches targeting HO-1 in chemoresistance have attracted great attention. In this review, we comprehensively summarized the impact of HO-1 on tumor chemoresistance, with a special focus on HO-1-mediated antioxidant, anti-apoptosis, and pro-autophagy activity, as well as proposed potential prospects of HO-1 inhibition in promoting therapeutic benefits in cancers.

## 2. Biological Functions of HO-1

HO-1, a member of the heme oxygenase (HO) family, also known as heat shock protein 32 (Hsp32), is encoded by the ubiquitous stress-responsive gene HMOX-1, which maps to chromosome 22q12.3 [44]. HO-1 is widely expressed as a type II membrane protein throughout mammals, which predominantly anchors to the endoplasmic reticulum membrane with its hydrophobic carboxy-termini. In addition, the distinct subcellular localization of HO-1 has also been reported, including the caveolae, the nucleus, and the mitochondria [45]. Under physiological conditions, normal cells express low levels of HO-1, which maintains cellular redox homeostasis [46]. Notably, HO-1 is highly inducible by various stress stimuli, including free heme, pathogen infection, tissue injury, altered oxygen tension (hypoxia/hyperoxia), inflammatory cytokines, UV radiation, and oxidant stress, which also serves as a protective feedback mechanism for cellular injury. As a stress response antioxidant enzyme, HO-1 was identified to play a critical role in heme metabolism by catabolizing toxic heme into free iron (Fe^2+^), carbon monoxide (CO), and the linear tetrapyrrole biliverdin, of which Fe^2+^ is sequestered by ferritin, and biliverdin is further reduced to bilirubin [47,48,49,50]. HO-1 is widely considered a remarkable cytoprotective factor and represents the cellular defense mechanism due to its enzymatic activity as well as byproducts, which will be further discussed below.

### 2.1. Degradation of Heme

Heme is synthesized in mitochondria by adding iron to protoporphyrin IX with ferrochelatase and transported to different cellular locations by membrane trafficking proteins as well as heme chaperones, which represent an indispensable cofactor for multiple biological processes, including oxygen binding and transport, electron transfer, oxidative metabolism, gas sensing, signal transduction related to apoptosis, and cell proliferation [51,52]. Intracellular heme homeostasis is tightly controlled because excess free heme is highly toxic by catalyzing the generation of free radicals, together with exacerbating lipid peroxidation, mitochondrial dysfunction, and inflammatory reactions [53,54]. Free heme triggers programmed cell death through various mechanisms. For instance, Fortes et al. indicated that heme induces macrophage necrosis through TLR4/Myd88-dependent expression of TNF and TLR4-independent generation of ROS [55]. Petrillo et al. found that heme accumulation in endothelial cells activates paraptosis by augmenting endoplasmic reticulum (ER) stress [56]. In addition, heme interacts with distinct immune cells to function as a proinflammatory molecule [57,58,59]. Heme itself is an activator of HO-1; in turn, the enzymatic activity of HO-1 eliminates excessive free heme, thereby alleviating cellular oxidative stress and cell injury. Hemin, an artificially synthesized form of heme, was demonstrated to cause DNA strand breaks and oxidative DNA damage in human colonic epithelial (HCEC) and colorectal cancer (CRC) cells, which subsequently promotes Nrf2 stabilization and translocation to the nucleus, along with robust expression of cytosolic HO-1, thereby conferring protection against the deleterious effects of hemin [60]. Nevertheless, a recent study described the opposite role of heme in HO-1 regulation: excessive concentrations of heme obviously compromised the inducibility of HO-1 in normal colonic epithelial cells [61]. Overall, degradation of free heme by HO-1 counteracts its cytotoxic effect, improving cellular homeostasis.

### 2.2. Metabolites of HO-1

#### 2.2.1. CO

CO is widely known as a poisonous gas due to its ability to bind to hemoglobin with great affinity and thereby block oxygen transport to vital organs, which can cause respiratory depression [62]. However, emerging evidence has suggested that CO, as a gasotransmitter, exerts antioxidant, anti-inflammatory, anti-apoptotic, anti-proliferative, anti-fibrotic, and anti-thrombotic properties at low to moderate doses [63,64,65,66]. Mitochondria are considered the central target organelle in the pro-survival action of CO [62,67,68,69]. The dominant function of mitochondria is ATP generation. Soluble guanylyl cyclase (sGC) is one of the most widely discerned molecular targets of CO [70]. CO elevates ATP production through the induction of sGC and inhibits TNF-α-mediated hepatocyte apoptosis [71,72]. In parallel, low concentrations of CO were found to protect astrocytes against oxidative stress-induced apoptosis by improving mitochondrial oxidative phosphorylation and ATP production, which relies on Bcl-2 expression and its interaction with cytochrome c oxidase (COX) [73]. CO was described to promote ATP production by activating AMPK phosphorylation, which coordinates with the activation of HIF-1α and Nrf2 to suppress hypoxia-induced tubular cell damage [74]. Additionally, CO was shown to increase the nuclear translocation of transcription factor EB (TFEB) via PERK-dependent Ca^2+^ signaling and calcineurin activation, which subsequently enhanced mitophagy and mitochondrial biogenesis [75]. Due to the great potential therapeutic value of CO, the development of carbon monoxide-releasing molecules (CORMs), which allow CO to be safely and selectively delivered, has recently attracted great attention [76].

#### 2.2.2. Fe^2+^

Iron is a vital element for numerous biological processes, including mitochondrial respiration, DNA synthesis, signal transduction, and oxygen transport. Iron is a transition element that can exist as reduced ferrous (Fe^2+^) or oxidized ferric (Fe^3+^). Fe^2+^ is implicated in the formation of cellular oxidative stress via the Fenton and Haber–Weiss reactions [77]. Concurrent with HO-1 activation, excessively produced Fe^2+^ is rapidly sequestered by ferritin in a redox-inactive form. Ferritin is a ubiquitously expressed cytosolic iron storage protein composed of two subunits, ferritin heavy chain (FTH) with ferroxidase activity and ferritin light chain (FTL), which facilitates iron nucleation [78].

Practically, ferritin can be regarded not only as an iron regulatory protein but also as a crucial cellular defense molecule against oxidative stress. In an in vitro oxidative stress model evoked by H_2_O_2_, robust and prolonged activation of JNK and p38 MAPKs was observed in a labile iron-dependent manner to trigger cell apoptosis. Along with this, ferritin showed a rapid elevation, which was considered a protective mechanism against external stimuli. Regrettably, the regulatory mechanism of iron in redox signaling was not fully elucidated in this study [79]. Another study showed that Nrf2, the master transcription factor involved in oxidative stress signaling modulation, may directly induce the transcription of the FTL and FTH genes [80]. Recently, ferritin was implied to interact with ferroptosis through selective autophagic degradation of ferritin via nuclear receptor coactivator 4 (NCOA4), namely, ferritinophagy [81,82], which sheds light on the treatment of various diseases, including cancers [83,84], neurodegenerative disorders [85], and immune dysfunction [86,87,88]. Ferroptosis exerts opposite therapeutic effects in a context-dependent manner; for instance, activating ferroptosis is beneficial for tumor therapy but has been found to promote the progression of neurodegenerative diseases. Thus, boosting or inhibiting ferroptosis reasonably in variable contexts to achieve satisfactory therapeutic effects might be possible.

#### 2.2.3. Biliverdin

Biliverdin (BV) is a water-soluble bile pigment that is rapidly converted to bilirubin (BR) by biliverdin reductase (BVR). The BV/BR redox cycle elicits robust defenses against oxidative stress and the inflammatory response. On the one hand, BV and BVR are well-characterized signaling cascades, and BVR exhibits cytoprotective effects by converting BV into the potent antioxidant BR. A previous study revealed that biliverdin reductase mRNA appeared to be increased by exogenous supplementation with biliverdin, which in turn reduced the infiltration of neutrophils and the expression of proinflammatory proteins in an HO-deficient corneal epithelial injury mouse model [89]. Similarly, recent studies have declared that the BVR-dependent BV/BR redox process protects lens epithelial cells (LECs) from oxidative stress-induced apoptosis by enhancing intracellular redox homeostasis, suppressing the NF-κB/iNOS pathway, and activating the Nrf2/HO-1 pathway [90]. On the other hand, BV itself evokes beneficial biological effects. BV could effectively diminish the upregulation of proinflammatory mediators such as IL-6, CCL2, and iNOS and relieve lipid peroxidation, giving rise to a reduction in intestinal mucosal injury in an in vivo intestinal transplantation model [91]. BV also inhibits cerebral ischemia reperfusion-induced cell apoptosis and ameliorates cerebral ischemia reperfusion injury (CIR) in the rat cortex by downregulating lncRNA H19 [92]. Another investigation proposed that elevated miRNA204-5p expression and its direct interaction with Ets1 was the underlying mechanism of BV-related neural improvement [93].

In addition, it is widely accepted that BR replaces BV to function as the final product of HO-1 metabolism in humans. Studies have provided evidence for the protective effects of BR via antioxidation, anti-inflammation, and immune modulation [94] in numerous diseases, including inflammatory bowel disease [95], diabetes [96], ischemia-reperfusion injury [97], and immunological diseases [98].

### 2.3. Subcellular Localization of HO-1

Apart from functions exerted by the enzyme activity, distinct subcellular compartments of HO-1 also affect its biological activity and function. The C-terminus of endoplasmic reticulum-located HO-1 can be proteolyzed under hypoxic or oxidative stress conditions, leading to different subcellular compartments of HO-1. However, it is somewhat surprising since HO-1 is inactive when translocated into the nucleus, whereas its enzyme activity is preserved when migrated to mitochondria or caveolae [45,99]. Nuclear HO-1 without enzymatic activity displayed the noncanonical function of HO-1, which may mediate oxidative stress protection and cell death prevention through the transcriptional regulation of antioxidant genes or signaling pathways, e.g., G6PDH, NQO1, PI3K/Akt/GSK3 [100], and MAPK pathways [101]. Furthermore, mitochondrial HO-1 was shown to protect lung epithelial cells from mitochondria-mediated cell death [102]. Moreover, upon mitochondrial translocation, HO-1 detoxified accumulated intramitochondrial free heme and reduced mitochondrial oxidative stress and dysfunction, which ultimately repaired apoptotic tissue injury in a gastric mucosal injury rat model [103].

Taken together, these findings imply that both increased expression or enhanced activity of HO-1 and the different subcellular locations of HO-1 variants constitute an important cellular protective mechanism against stressful stimuli.

## 3. Mechanisms of HO-1 Regulation

### 3.1. Transcription Factors

The human HO-1 gene (HMOX-1) contains specific DNA-binding elements in the promoter, namely, the proximal promoter (PP), distal enhancer E1 (DE1), and distal enhancer E2 (DE2), located at approximately −0.3 kb, −4 kb, and −10 kb, respectively [104,105,106]. Accumulating evidence has revealed that multiple transcription factors may directly bind to enhancer sequences termed “stress-related response elements” (StREs) in the promoter region of HMOX1, which regulates the expression of HO-1 [106].

Nuclear factor erythroid derived 2-like 2 (Nrf2) and BTB and CNC homology 1 (BACH1) are considered the key transcription factors that conversely modulate HO-1 expression (Figure 1). Nrf2 and BACH1 are both pivotal defense molecules against oxidative stress that bind to similar DNA sequences by forming respective heterodimers with one or more members of the small Maf (musculoaponeurotic fibrosarcoma) protein family. Under basal conditions, Nrf2 is sequestered in the cytoplasm by Kelch-like ECH-associated protein 1 (Keap1), leading to the ubiquitination and subsequent proteolysis of Nrf2, which retains the low activity of Nrf2 [107]. In response to oxidative stress, Nrf2 dissociates from Keap1 and translocates to the nucleus. Then, through the basic leucine zipper (bZip) domain, Nrf2 heterodimerizes with Maf family proteins to bind StRE or ARE (antioxidant response elements) of the target genes [108,109]. It is through this mechanism that Nrf2 positively regulates HO-1 expression under stimulus conditions. In contrast to Nrf2, under normal conditions, BACH1 is stably expressed in the cytosol and transferred into the nucleus to bind to Maf proteins as a transcriptional repressor, which blocks the nuclear translocation of Nrf2 and consequently represses the expression of HO-1 [110]. Under high heme concentrations, BACH1 dissociates from Maf proteins, is exported from the nucleus, and is degraded after tyrosine phosphorylation. Then, Nrf2 directly binds to the HO-1 promoter, leading to the activation of HO-1 expression [111]. Moreover, existing studies also suggest that HMOX1 induction can be BACH1 dependent but NRF2 independent [112,113].

In addition to Nrf2 and BACH1, transcription factors such as HIF1α, NF-κB, ATF4, AP-1, and KLF7 also regulate HO-1 at the transcriptional level by converging on the HO-1 promoter, among which HIF1α [114,115], NF-κB [116], ATF4 [117], and AP-1 [118] positively regulate HO-1 expression, while KLF7 [119] is known as a repressor of HO-1. Of particular interest, HIF1α, a key regulator of hypoxia-stimulated metabolic adaptation, has been considered to be an activator of HO-1 by multiple in vitro and in vivo studies, which has attracted research attention. It was implied in a recent study that HO-1 acts both downstream and upstream of HIF-1α, and HO-1-stabilized HIF-1α may partly be due to its enzymatic activity [120]. The interaction between the two stress adaptation molecules is sophisticated, and its impact on physiological and pathological states is worthy of further study.

### 3.2. Promoter Polymorphisms

Gene polymorphisms located in the promoter region are thought to affect gene transcription. In the 5’-noncoding region of the HMOX-1 gene, three polymorphic sites were identified to be functional and regulate HO-1 expression, including the (GT)n repeat dinucleotide length polymorphism and T(−413)A and G(−1135)A single nucleotide polymorphism (SNP) sites [121,122]. Among these, the (GT)n polymorphism was the most studied. Current evidence proved that the (GT)n repeat varies from 12–45 repeats [123], and a GT repeat sequence < 25 was defined as a short (S) allele, relating to higher HO-1 inducibility and enhanced HO-1 enzymatic activity compared to those with ≥25 repeat sequences, which were considered long (L) alleles [124]. The other two single-nucleotide polymorphisms, T(−413)A and G(−1135)A, were detected in research aimed at exploring the correlation between HO-1 promoter variants and human essential hypertension. It was shown that the A(−413) allele was associated with significantly improved HMOX-1 promoter activity in comparison with the T(−413) allele [125]. A previous study comprehensively considered both T(−413)A and (GT)n polymorphisms to investigate the promoter activity of HMOX-1, and the results revealed that the transcriptional activity of the A(−413)-(GT)30 allele was approximately six times higher than that of the T(−413)-(GT)23 allele [126]. However, the effect of the G(−1135)A polymorphism on HO-1 transcriptional activity remains unclear.

### 3.3. MicroRNAs

Emerging evidence indicates that microRNAs are involved in the transcriptional and posttranscriptional modulation of the HO-1 gene, which opens up a new horizon for research on HO-1 regulation. For instance, it was demonstrated that miR-155 efficiently upregulated HMOX1 mRNA and protein expression by inhibiting BACH1 protein translation in human umbilical vein endothelial cells [127]. Gu et al. also found that miR-155 positively regulated HO-1, which favors lung cancer resistance to arsenic trioxide [128]. In contrast to the aforementioned results, Zhang et al. reported that miR-155 could directly target the 3′UTR of HO-1, which repressed its expression in tolerant CD4+ T cells [129]. In addition, Li and colleagues found that miR-155 and miR-181a synergistically engaged in cadmium-induced kidney immunotoxicological effects by reducing HO-1 expression [130]. miR-494 was suggested to upregulate HO-1 expression through a BACH1-independent mechanism in neuroblastoma cells under oxidative stress [131]. miR-1254 and miR-193a-5p were shown to be HO-1 suppressors and inducers in prostate and non-small cell lung cancer cells, respectively [132,133]. Moreover, microRNAs may modulate HO-1 expression at a posttranslational level, as it was proposed that overexpression of miR-217 combined with miR-377 decreased HO-1 protein expression but did not change HMOX1 mRNA levels [134].

## 4. Cytoprotective Role of HO-1 in Cancer

The primary roles of HO-1 in antioxidant, anti-inflammatory, antiapoptotic, and proangiogenic effects have been highlighted [135]. Nevertheless, HO-1 displays equivalent cytoprotective effects on tumor cells against oxidative stress caused by the accumulation of ROS [136]. Thus, HO-1 has been implicated in a broad spectrum of protumorigenic effects and various cancer hallmarks. Survival analysis in the PrognoScan database (www.prognoscan.org/, accessed on 10 March 2023) provided strong evidence that high HO-1 mRNA expression predicts significantly worse prognosis in lung cancer, breast cancer, blood cancer, brain cancer, colorectal cancer, and ovarian cancer (Figure 2). Meanwhile, in the Cancer Treatment Response Gene Signature Database (CTR-DB, http://ctrdb.cloudna.cn/) [137], the HMOX-1 expression level was shown to be upregulated in treatment-nonresponsive patients with breast and brain cancer compared with responsive patients (Figure 3).

HO-1 overexpression drives tumor growth, metastasis, angiogenesis, therapy resistance, and immune evasion. Wang et al. found that GRIM-19 deficiency resulted in aberrant HO-1 activation in a ROS-Nrf2 axis-dependent manner in gastric cancer cells, along with significantly increased HO-1 expression in metastatic lung and liver tissues, while HO-1 inhibition limited GC cell migration and invasion and directly abrogated GC metastasis in vivo [138]. Overexpressed HO-1 also augmented the bone metastasis of prostate cancer by modulating bone turnover and remodeling. More importantly, HO-1 enhanced the cell-cell interactions of osteoblasts and cancer cells [139]. A more recent study illustrated that exosomes may serve as the basis for HO-1-involved intracellular communication because androgen-independent prostate cancer (AIPC) cell-derived exosomes activated HMOX1 expression in androgen-dependent prostate cancer (ADPC) cells in vivo and in vitro, which promoted the transformation of ADPC cells to AIPC cells and the development of castration-resistant prostate cancer [140]. In addition, HO-1-related angiogenesis plays an essential role in the progression of solid tumors, and the underlying mechanism is likely due to its upregulation or activation of proangiogenic factors such as VEGF and stroma cell-derived factor-1 (SDF-1) [141,142,143].

Notably, an increasing amount of evidence suggests that HO-1 acts as an immunosuppressive agent in immune responses [144,145]. HO-1 expression was induced during the differentiation of monocytic cells into macrophages in the tumor microenvironment (TME) and exerted a strong immunosuppressive effect by limiting antigen-specific CD8+ T-cell effector function against tumor cells, while myeloid-restricted HO-1 ablation boosted the effectiveness of therapeutic immunization [146]. Consistently, Khojandi et al. revealed that HO-1 endowed tumors with the ability to resist immune-mediated apoptosis, and combined treatment with HO-1 inhibition and anti-PD1 significantly decreased tumor volume in a mouse model of breast cancer and melanoma [147]. Indeed, HO-1 may affect the immune response in many distinct ways, among which two main approaches should be highlighted. On the one hand, HO-1 expressed by antigen-presenting cells (APCs) interferes with the development of regulatory T (Treg) cells, along with restricted infiltration and activity of effector T (Teff) cells, which was considered as the fundamental mechanism of the HO-1-related immunosuppressive phenotype [148,149,150]. On the other hand, the promotion of HO-1 expression is associated with biased M2 macrophage polarization, thus facilitating suppressed immune responses and enhancing tumor progression [151,152]. Furthermore, the inhibition of HO-1 augments tumor cell recognition and elimination by NK cells [153,154]. Therefore, HO-1 represents a promising target to reprogram the TME and improve the efficacy of cancer immunotherapy.

Based on their promising therapeutic potential, the development of HO-1 inhibitors has attracted increasing attention. The most reported HO-1 inhibitors for experimental use are metalloporphyrines (Mps), known as heme derivatives, including SnPP, SnMP, and ZnPP, which are structurally similar to protoporphyrin but have different metal ions in their center. Mps can competitively inhibit heme binding to HO-1 to decrease the enzyme activity. However, the clinical application of Mps is limited due to the inhibition of other heme-containing enzymes with poor selectivity, which largely limits its clinical translation [155,156]. Azole-based derivatives such as ketoconazole, terconazole, sulconazole nitrate, and imidazole-based compounds used for HO-1 inhibitors are also reported (Figure 4) [157,158]. Ketoconazole, as a widely used antifungal agent, was indicated to inhibit both HO-1 and HO-2 activity at typical therapeutic concentrations, and the potential mechanism is that the KTZ imidazole moiety directly interacts with heme iron and forms a complex blocking heme from binding to the HO catalytic site [158]. Aside from its antifungal activity, ketoconazole has been applied off-label as a second-line hormonal therapy agent for castration-resistant prostate cancer (CRPC) since the 1980s, due to its additional antiandrogenic function via inhibition of CYP17A1 to block androgen synthesis. Numerous studies have demonstrated that ketoconazole significantly decreases PSA levels, alleviates clinical symptoms, and delays disease progression in CRPC patients, regardless of whether it is used in combination with docetaxel or applied alone before and after docetaxel [159,160]. Furthermore, artesunate, a powerful anti-malaria drug, was declared to have a potential impact on HO-1 modulation [161], iron metabolism [162], and ferroptosis [163]. Concurrently, another approved drug, artesunate, was announced to be active against cancers, including ovarian cancer [164], hepatocellular carcinoma [165], renal cell carcinoma [166], breast cancer [167], and colorectal cancer [168]. Although these drugs showed both HO-1 modulation and cancer therapy effects, whether the HO inhibitory effect directly contributes to the effectiveness of tumor therapy remains unclear.

Conspicuously, aberrant activation of HO-1 was universal in chemoresistant cancer cells compared to that of chemosensitive cancer cells, while the inhibition of HO-1 obviously mitigated resistance to anticancer therapies. These results suggest that HO-1 is implicated in the chemoresistance of various human cancers and could be a potential therapeutic target to overcome chemotherapy failure. This aspect will be discussed in the next section.

## 5. Mechanisms of HO-1-Targeted Chemoresistance in Cancer

Chemotherapeutic agents related ROS accumulation can eliminate cancer cells by inducing oxidative stress, causing DNA damage, disrupting mitochondrial function, and facilitating synergistic effects [9]. However, its efficacy and long-term use are seriously limited by drug resistance. Recently, a pivotal role of HO-1, a cytoprotective enzyme with known antioxidant defense functions, in chemoresistance has been highlighted. The upregulation of HO-1 after chemotherapy in cancer cells can mitigate ROS-mediated oxidative damage and counteract chemotherapeutic agent-induced cytotoxicity, which contributes to acquired chemoresistance. It has also been well documented that the inhibition of HO-1 is capable of restoring the chemosensitivity of resistant cells in multiple cancer types, including ovarian cancer [169], acute myeloid leukemia [170], and melanoma [154].

To date, the underlying mechanisms of how HO-1 promotes chemoresistance have not been fully elucidated. An increasing number of studies have suggested that antioxidative, anti-apoptosis, and pro-autophagy activities might be involved in the development of HO-1-dependent chemoresistance. In this section, we will comprehensively summarize the role of these mechanisms in tumor chemoresistance associated with HO-1.

### 5.1. Antioxidative Activity of HO-1

As described above, induction of intracellular ROS by conventional chemotherapeutic agents can be beneficial in terms of their cytotoxic effects on cancer cell. Yet, enhanced HO-1 expression and enzymatic activity protect cancer cells from ROS during stress response, leading to acquired drug resistance. Sun et al. reported that the Nrf2/HO-1 pathway positively regulated by SIRT5 contributed to cisplatin resistance in ovarian cancer cells by suppressing cisplatin-induced DNA damage in an ROS-dependent manner [169]. The upregulation of Nrf2 and its downstream target HO-1 has also been reported to be correlated with 5-Fu resistance in gastric adenocarcinoma by inhibiting ROS production [171]. These observations broadly coincide with the results of our previous work. We found that Nrf2 and its target genes, including HO-1 and NQO-1, were upregulated in cisplatin-resistant ovarian cancer cells, and Nrf2, the main regulator of HO-1, was of great importance for the development of cisplatin resistance in ovarian cancer [172,173,174]. In addition, while induction of HO-1 expression by hemin was shown to protect laryngeal squamous cancer cells from cisplatin-induced oxidative damage and apoptosis in vitro, the suppression of HO-1 expression and enzyme activity by the HO-1 inhibitor ZnPPIX significantly promoted ROS downstream signaling pathway activation, such as P38 and JNK phosphorylation, which conclusively enhanced the cisplatin sensitivity of cancer cells [175].

The mechanisms underlying chemoresistance are also involved in the antioxidant properties of heme metabolites catalyzed by HO-1. Recently, Rios-Arrabal et al. reported that CO produced after HO-1 overexpression induced ECE-1 expression through the activation of pNF-kβ and pc-Jun in p53 wild-type colorectal cancer (CRC) cells, leading to 5-FU resistance. However, CO released by HO-1 was unable to modify 5-FU sensitivity in P53 null CRC cells, suggesting that CO-involved chemoresistance may partly rely on P53 [176]. Although HO-1-produced free ferrous iron itself is considered a mediator of ferroptosis [177], a new type of cell death mediated by iron-dependent ROS and lipid peroxidation, free ferrous iron can be rapidly sequestered and stored by ferritin. The latter has been well documented to have a significant impact on chemoresistance. For instance, ferritin was found to prevent doxorubicin-mediated cell death by inhibiting intracellular ROS formation and then reduce doxorubicin sensitivity in a dose- and time-dependent manner in breast cancer cells [178]. Salatino A et al. observed that chemoresistant ovarian cancer patients may be characterized by higher ferritin heavy chain levels than chemosensitive patients, and overexpression of ferritin heavy chain significantly eliminated cisplatin-mediated ROS, subsequently leading to reduced responsiveness in ovarian cancer cells [179]. Indeed, studies on the association between the cytoprotective role of HO-1 and newly identified forms of cell death, such as ferroptosis [180], necroptosis [181], and pyroptosis [182], are gradually emerging; however, the role of this linkage in chemoresistance remains poorly defined.

In conclusion, these data implied that the antioxidative and cytoprotective features of HO-1 and its byproducts’ attributes to the development of chemoresistance and poor patient prognosis. However, evidence has not been provided that HO-1 possess direct ROS-scavenging activity like “classical” antioxidant enzymes (SOD, catalase, peroxiredoxins, etc.). Instead, HO-1 exert antioxidant effects indirectly, and the possible mechanism includes its removal of “prooxidant” heme and production of antioxidant biliverdin/bilirubin and/or CO products. Moreover, co-expression of HO-1 with other protective enzymes may obscure the real driving force of the antioxidant effect.

Meanwhile, emerging studies have revealed that HO-1 is a double-edged sword in ROS regulation; when excessively activated, HO-1 can induce ROS overload and cancer cell death [183,184]. The probable mechanism may involve uncoordinated stress response mechanisms mainly caused by imbalanced iron metabolism. Some scholars have concluded that the iron deprivation status of the body impels the antioxidant function of HO-1 [185]. Since HO-1 regulates iron metabolism via the degradation of heme, once excessively produced iron cannot be utilized for biosynthesis or sequestered by ferritin, oxidative stress occurs, which also initiates ferroptotic cell death [186,187]. Hence, the dual role of HO-1 in ROS modulation potentially substantiates its paradoxical action in ferroptosis.

### 5.2. Anti-Apoptosis Activity of HO-1

Apoptotic cell death, a form of programmed cell death provoked in tumor cells following exposure to chemotherapeutic agents and excessive oxidative stress [188,189], is integral to the success of cancer chemotherapy. Since enforced expression and activation of HO-1 in response to chemotherapy-mediated oxidative stress could provide cytoprotective effects against chemotherapy-induced apoptosis in tumor cells, this antiapoptotic mechanism might contribute to acquired chemoresistance. A protein-protein interaction (PPI) network of HMOX1-related genes was constructed using the STRING database (Figure 5A). To further explore the function of HMOX1, these interacting genes from the PPI network were used for Gene Ontology (GO) enrichment analysis. The GO analysis results showed that the HMOX1-related genes were mainly enriched in antioxidative stress and anti-apoptosis biological processes (Figure 5B). Based on this, we will fully address the influence of HO-1 on apoptosis, apoptosis-related proteins, and signaling pathways in resistant tumor cells.

The transcriptional and protein levels of pro-apoptosis-related genes, such as Bax, Smac, Survivin, Fas/FasL, caspase-3 or caspase-9, and anti-apoptosis genes, such as Bcl-2, BFAR, Bcl-xL, A1 or Mcl-1, are often measured to estimate the degree of cellular apoptosis [190,191]. Barbagallo et al. described that increased HO-1 gene expression and enzymatic activity drive resistance to carfilzomib in neuroblastoma cells, while cotreatment with carfilzomib and LS1/71, a noncompetitive inhibitor of HO-1, triggered cell apoptosis with significantly activated proapoptotic BAX gene expression and reduced antiapoptotic BFAR gene expression [192]. Chakraborty et al. determined that the cytoprotective effect of the c-Met-HGF-Nrf2-HO-1 pathway in sorafenib-treated renal cancer cells was mainly implicated in the reduction in ROS generation, promotion of Bcl-2 and Bcl-xL expression, and downregulation of cleaved caspase-3 expression, which ultimately led to apoptosis resistance [193]. These results are consistent with the observation that HO-1 inhibition sensitized pancreatic ductal adenocarcinoma cells to gemcitabine by increasing the production of reactive oxygen species, disrupting the glutathione cycle, and enhancing apoptosis [194]. Recently, microRNAs involved in the regulation of HO-1 were suggested to participate in the development of chemoresistance by decreasing apoptosis. Gu et al. confirmed that HO-1 expression modulated by miR-155 was partly responsible for the downregulation of arsenic trioxide-induced apoptosis, leading to resistance to arsenic trioxide in lung cancer cells [128]. miR-193a-5p-induced HO-1 expression was shown to counteract docetaxel-induced apoptosis in PC3 cells [133].

In addition, considerable research efforts have been devoted to investigating the impact of HO-1 end products generated by heme metabolism on apoptosis evasion. Huang et al. proposed that activation of the HO-1/CO axis protects lens epithelial cells (LECs) from H_2_O_2_-induced apoptosis by inhibiting oxidative stress and activating NF-κB signaling, along with decreasing apoptotic molecules (Bax, Bcl-2, and caspase-3) [195]. A recent in vitro study revealed that CO produced by HO-1 may contribute to the resistance of myeloma cells to bortezomib. Specifically, the treatment of myeloma cells with bortezomib induced cross-regulation between TLR4 and the HO-1/CO signaling pathway, which subsequently increased the unfolded protein response and protected the function of mitochondria, leading to decreased apoptosis resulting from the cytotoxic effects of bortezomib [196]. In fact, it should be noted that existing studies have mainly focused on the apoptosis evasion role of HO-1 itself in chemoresistance, and further effort is still required to understand the antiapoptotic mechanisms regulated by HO-1 byproducts in chemoresistance.

### 5.3. Pro-Autophagy Activity of HO-1

Autophagy, an evolutionarily conserved catabolic process, plays an essential role in maintaining cellular homeostasis under various metabolic stresses, such as nutritional deficiency, hypoxia, pathogen infection, ER stress, oxidative stress, and even chemotherapy intervention, by capturing and transporting aggregated or misfolded proteins and damaged organelles to lysosomes for degradation [197,198]. To date, three major types have been identified, including macroautophagy, chaperone-mediated autophagy, and microautophagy. Of these, macroautophagy (hereafter autophagy) is the most studied.

In cancer cells, autophagy is a double-edged sword for cell fate determination. On the one hand, excessive or persistent autophagy promotes autophagic cell death, limits tumorigenesis and enhances therapeutic efficacy in some specific contexts [199]. On the other hand, the prosurvival function of autophagy is thought to be involved in oncogenesis, tumor progression, and therapy resistance [200].

HO-1 and autophagy are both inducible by stress conditions. Indeed, HO-1 was revealed to be coregulated with autophagy in the development of tumor chemoresistance, and manipulating HO-1-related autophagy to improve therapeutic responses has attracted attention. A previous study showed that HO-1 expression was upregulated by rapamycin and sorafenib treatment in renal cancer cells, which subsequently promoted cancer cell survival by suppressing both apoptotic and autophagic cell death. The underlying mechanism of HO-1-downregulated autophagy was associated with increased linkage between Beclin-1 and Rubicon [201]. Nevertheless, the positive role of HO-1 in autophagy activation to promote cell survival has been highlighted. HO-1 activated autophagic flux by increasing the LC3BI/LC3BII ratio and upregulating ERK and JNK, which was indicated to drive resistance against HER2-targeted therapies in breast cancer [202]. HO-1 boosted autophagy modulation by the Src-STAT3 and PI3K/Akt signaling pathways and was also discovered to contribute to doxorubicin and pharmorubicin resistance, respectively, in breast cancer cells [42,43]. In chronic myeloid leukemia cells, silencing HO-1 was shown to reverse imatinib resistance by inhibiting autophagy through the activation of the mTOR pathway [203]. In addition, increased HO-1 induced by silencing BACH2 was demonstrated to facilitate bortezomib resistance in mantle cell lymphoma cells by triggering cytoprotective autophagy formation and maintaining ROS at a minimal tumor-promoting level [204].

The underlying mechanism of HO-1-regulated autophagy is complicated and not fully elucidated. Autophagy and apoptosis are tightly intertwined, and the regulation of autophagy by HO-1 is often accompanied by changes in apoptosis, either by inducing autophagy to inhibit apoptosis and play a protective role [205,206,207] or activating excessive autophagy to blow up apoptotic cell death [208]. Thus, appropriate regulation of HO-1-associated autophagy might become a potential therapeutic strategy to ameliorate chemoresistance. It is noteworthy that mitochondria are critical to the manipulation of apoptosis, energy production and cellular homeostasis [209]. Mitophagy, a selective form of autophagy that facilitates the elimination of damaged mitochondria before they cause cellular function impairments, can be amplified by overactivation of HO-1 and then undergo cell demise in glioma cells [210]. In accordance with this, HO-1-dependent NRF-1 induction was validated to arrest cardiomyocyte cell death and prevent fibrosis after oxidative stress through the transcriptional regulation of key mitophagy proteins that preserve mitochondrial homeostasis [211]. In addition, AMPK, a crucial regulator of energy metabolic homeostasis, was recently found to antagonize mTOR and facilitate HO-1-associated autophagy [212]. HO-1 was implicated in maintaining mitochondrial quality by inhibiting mitochondrial fission [213,214]. Based on the evidence reviewed above, the crosstalk between autophagy and apoptosis as well as the participation of mitochondria in biological processes might be relevant to the regulation of autophagy by HO-1, which still remains to be thoroughly investigated.

## 6. Conclusions

HO-1 is an established component of the antioxidant defense system that exerts cytoprotective functions by scavenging ROS and recovering redox homeostasis to counteract oxidative stress. However, recent developments have suggested that aberrant activation of HO-1 may be the decisive factor for chemoresistance in various cancers. In this review, we highlighted that increased antioxidative, antiapoptotic, and proautophagic activities are crucial mechanisms underlying HO-1-mediated chemoresistance (Figure 6).

Research to date has revealed that inhibition of HO-1 may be a promising strategy to overcome chemoresistance and improve the efficacy of chemotherapeutic regimens. Both pharmacological suppression of HO-1 activity by zinc protoporphyrin IX (ZnPP-IX) or tin protoporphyrin IX (SnPP-IX) and targeted knockdown of HO-1 expression by siRNA significantly enhanced chemosensitivity in cancer cells and suppressed chemoresistant xenograft tumor growth in vivo. Advances have been made in understanding the mechanisms underlying HO-1-mediated chemoresistance, and further development of HO-1-targeted therapies provides encouraging prospects for clinical application to optimize patient prognosis in different cancers. However, translating preclinical results to the clinic has been challenging due to the lack of safe and effective HO-1 inhibitors. It is noteworthy that suppression of HO-1 might break its protective effect on normal cells when exerting a killing effect on tumor cells; thus, the side effects of HO-1 inhibitors on normal cells and the incidence of adverse outcomes remain to be properly measured in the development of new inhibitors. Furthermore, whether HO-1-dependent therapy resistance is universal or specific for certain tumor types remains largely unexplored. As discussed above, overexpressed HO-1 leads to therapy resistance in both solid and hematological tumors. As a future perspective, the detailed molecular mechanisms of how HO-1 meditates chemoresistance in a specific cancer type and the development of safe, effective HO-1 inhibitors warrant further investigation.

## Figures and Tables

**Figure 1 antioxidants-12-01217-f001:**
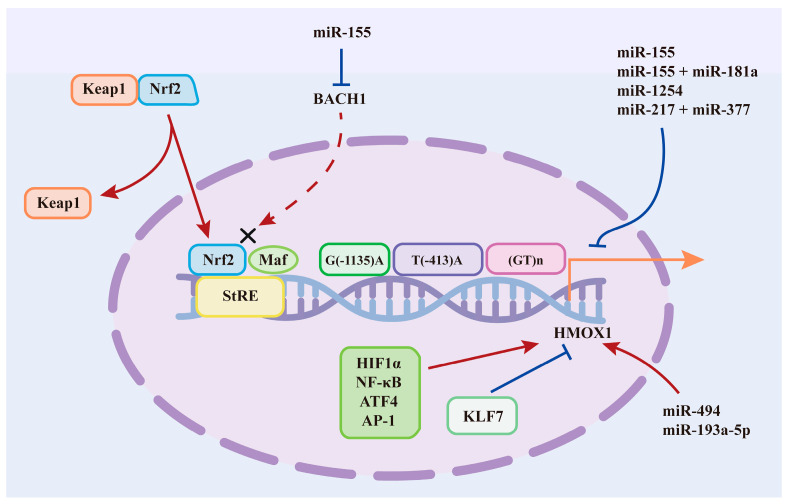
The regulatory mechanism of HO-1. Upon stressful stimuli, Nrf2 dissociates form the Keap1–Nrf2 complex and translocates into nucleus, and then Nrf2 binds with small Maf proteins to stress response elements (StRE) in the regulatory regions of HMOX1, leading to the activation of HO-1. BACH1 acts as a negative regulator of Nrf2 through competing with Nrf2 for binding to Maf proteins, which inhibits HO-1 expression. Several other important transcription factors are also implicated in the modulation of HO-1 expression including HIF1α [114,115], NF-κB [116], ATF4 [117], AP-1 [118], and KLF7 [119]. Among these transcription factors, HIF1α demonstrate a complex crosstalk with HO-1 [120]. Genetic polymorphisms in the promoter region of HMOX1 including (GT)n repeat and T(−413)A SNP are also involved in the transcriptional regulation HO-1 [121,122,123,124,125,126]. MiR-155 displayed opposite roles in HO-1 regulation. On the one hand, miR-155 activates HO-1 expression partially via repressing BACH1 [127,128], while on the other hand, miR-155 alone or in cooperation with miR-181a inhibits HO-1 expression [129,130]. Furthermore, it has been reported that miR-494 and miR-193a-5p enhanced HO-1 expression, wihle miR-1254 and miR-217 combined with miR-377 suppress the activity of HO-1 [131,132,133,134].

**Figure 2 antioxidants-12-01217-f002:**
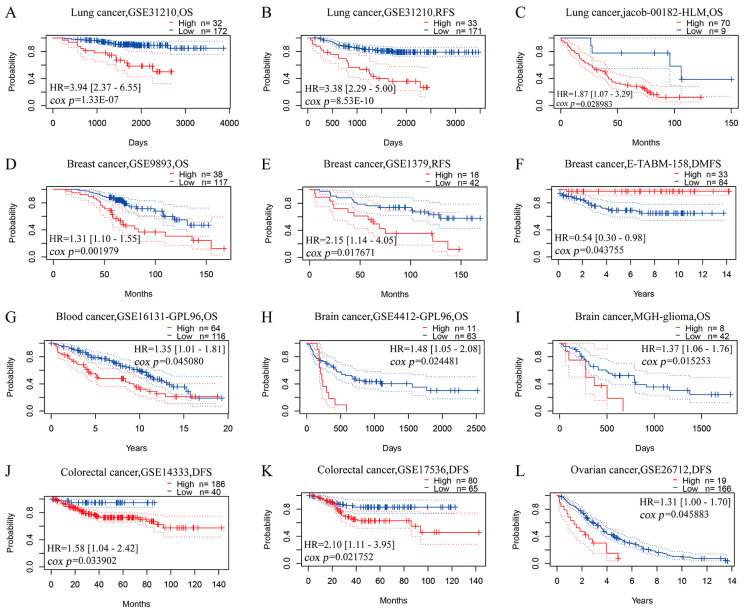
Prognostic significance of HMOX1 expression in various cancer types. Kaplan-Meier survival curves obtained from PrognoScan database for patients with high (red) and low (blue) HO-1 mRNA expression in lung cancer (**A**–**C**), breast cancer (**D**–**F**), blood cancer (**G**), brain cancer (**H**,**I**), colorectal cancer (**J**,**K**), and ovarian cancer (**L**). OS, overall survival; RFS, relapse free survival; DMFS, distant metastasis free survival; DFS, disease free survival; HR, hazard ratio.

**Figure 3 antioxidants-12-01217-f003:**
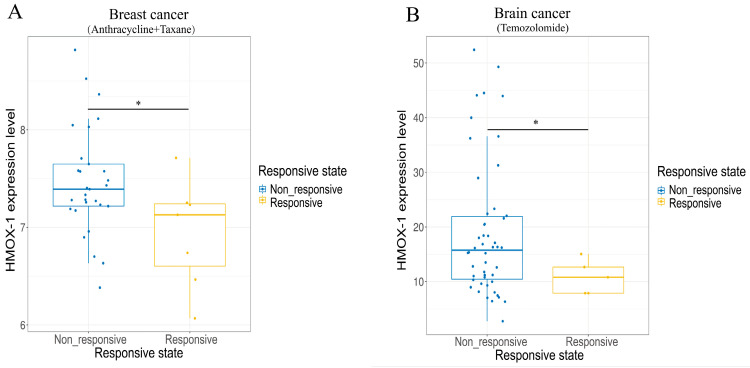
HMOX-1 expression differences between non-responsive and responsive groups in breast (**A**) and brain cancer (**B**) during chemotherapy. * *p* < 0.05.

**Figure 4 antioxidants-12-01217-f004:**
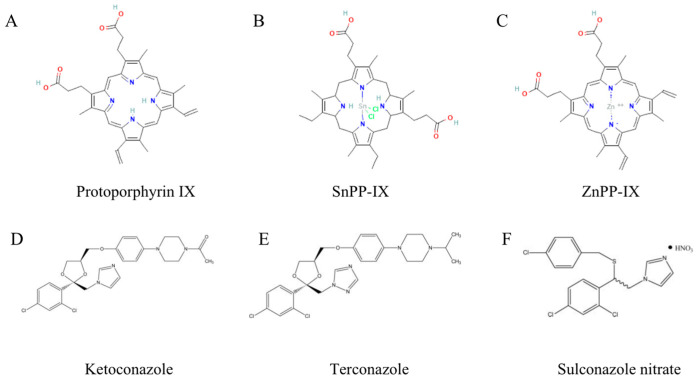
Chemical structure of mainstream HO-1 inhibitors. (**A**–**C**). Protoporphyrin class (cited from PubChem). (**D**–**F**). Azole-based derivatives [158].

**Figure 5 antioxidants-12-01217-f005:**
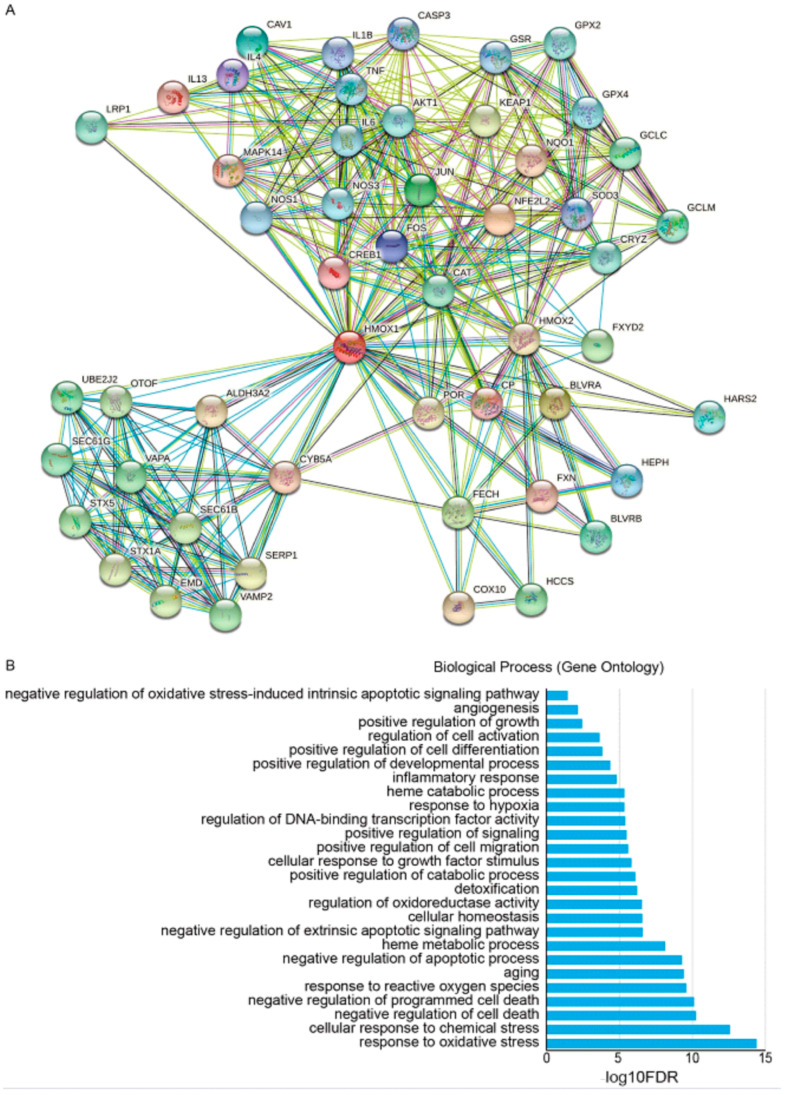
Protein-protein interaction network of HMOX1-related genes associated with antioxidative stress and anti-apoptosis factors. (**A**) The protein-protein interaction network of HMOX1-related genes based on STRING database (https://www.string-db.org/). (**B**) Gene Ontology (GO) enrichment analysis of HMOX1-related genes in protein-protein interaction network categorized by biological process.

**Figure 6 antioxidants-12-01217-f006:**
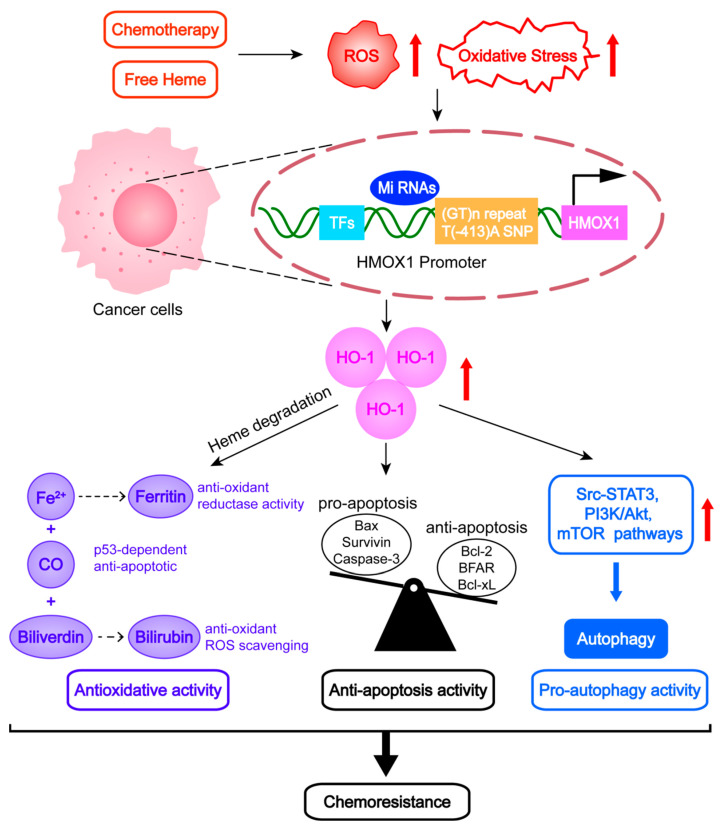
Schematic representation of heme oxygenase 1 (HO-1) regulation and the potential mechanism of HO-1-mediated chemoresistance.

## Data Availability

All data generated or analyzed during this study are included in this published article.

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
