# Peer review of "Cytoprotective Role of Heme Oxygenase-1 in Cancer Chemoresistance: Focus on Antioxidant, Antiapoptotic, and Pro-Autophagy Properties"

_antioxidants, 2023, doi:10.3390/antiox12061217_

Round 1
Reviewer 1 Report
The authors review the role of heme oxygenase 1 in chemoresistance. This may be of interest to the general journal audience. There are some minor concerns and editorial errors, which should be addressed by the authors, as listed below:
1. Throughout the manuscript, the authors suggest that HO-1 possesses ROS-scavenging activity. In contrast to “classical” antioxidant enzymes (SOD, catalase peroxiredoxins, etc.), HO-1 does not show any direct ROS-scavenging activity and its antioxidant effects are indirect and may be due to: (i) removal of “prooxidant” hemes; and (ii) production of antioxidant biliverdin/bilirubin and/or CO products. In addition, HO-1 expression is typically induced in parallel with other protective enzymes, including antioxidant enzymes. This may result in misinterpretation of the experimental data and attributing antioxidant effects of co-expressed enzymes to HO-1. It is, therefore, important that the authors clearly state what antioxidant properties HO-1 possesses, and which ones it does not.
2. While ROS may contribute to the cytotoxic effects of chemotherapeutics, they have other mechanisms that may be more important (DNA alkylation, inhibition of DNA repair machinery, TKI inhibition, etc.). Therefore, it would be more accurate to state that ROS may contribute to the cytotoxic effects rather than claiming that ROS formation is the principal mechanism of chemotherapeutics.
3. Protoporphyrins (ZnPP-IX, SnPP-IX) are not the only class of HO-1 inhibitors. It would be beneficial for the readers if the authors include a paragraph on reported small molecule HO-1 inhibitors, including the chemical structures.
4. The accuracy of the references need improvement. For example, ref. 3 does not focus on ROS in cancer treatment. Adding references to some reviews of the role of ROS in cancer development and treatment would be beneficial for the readers.
5. Characters representing the radical character (dots) and charge should be used as a superscript.
6. Line 37: change “orbit” to “orbital”; “superoxide” to “superoxide radical anion”; “hydroxyl” to “hydroxyl radical”
7. Line 39: remove “hydroxide anion” as it does not belong to ROS
8. Line 40: format “2” as a subscript in singlet oxygen (1O2)
9. Line 40: the ability to be transformed to a radical is not the reason to assign a molecule to ROS. For example, water can be transformed into a hydroxyl radical, but is not regarded as “ROS”. Oxidizing potential of the downstream products of incomplete reduction of molecular oxygen seem to be a more accurate description.
10. Line 42: what is the meaning of the term “delicate”? Consider a change to a more specific description.
11. Lines 43-44: the authors missed a class of antioxidant enzymes, peroxiredoxins, which are the major ‘players” in the removal and signaling by hydrogen peroxide.
12. Line 44: the sentence suggests that glutathione belongs to antioxidant enzymes, which is not the case.
13. Line 82: change “antioxidant molecule” to “antioxidant enzyme”
14. Line 126: change “locales” to “locations” or “sites”
15. Line 198: change “pigmen” to “pigment”
16. Using Supplementary data containing a single figure should be avoided in a review paper. If the authors believe this figure is important, it should be included in the main text.
Some minor error need to be fixed.
Author Response
Thank you very much for the kind evaluation and all the valuable comments. We made our every effort to address all the concerns one-by-one below and the responses were uploaded as "Responses to Reviewer 1". We hope you will be satisfied with the revised manuscript.

Reviewer 2 Report
In this review the authors summarized the function of the heme oxygenase-1 (HO-1) molecules in cancer with a specific focus on cancer chemoresistance and the underlying moleculr mechanisms.
It is a well written and timely review that encompasses all the major mechanisms of HO-1 regulation.
Also the figures are well organized and sufficient for the comprehension of the text.
Author Response
Thank you very much for the positive comments and encouragement.
Reviewer 3 Report
The manuscript by Kaikai Chang, Xiaofang Yi and collaborators is an interesting and comprehensive review of the role that the Heme-Oxygenase 1 (HO-1) protein plays at a physiological and pathological level. The review is very well written and well organized and also provides information on the most recent discoveries related to HO-1.
I therefore believe that the manuscript can be accepted for publication after making some small changes:
- On line 113 the bibliographic references are given between () rather than [].
- The sentence starting on line 125 is too long and it would be better to split it into several sentences
- The sentence in line 223 is unclear, it should be rewritten
- The resolution of figure 2 has to be enhanced
- In the legend of figure 3 it would be advisable to include the relative bibliographic reference, since it cannot be deduced from the main text
- The sentence starting on line 125 is too long and it would be better to split it into several sentences
- The sentence in line 223 is unclear, it should be rewritten
Author Response
Thank you very much for the kind evaluation and all the valuable comments. We made our every effort to address all the concerns one-by-one below and the responses were uploaded as "Responses to Reviewer 3". We hope you will be satisfied with the revised manuscript.

Reviewer 4 Report
The authors reviewed the cytoprotective role of heme oxygenase-1 in patients with cancer. This paper is well reviewed and written. However, I have some minor comments.
Comments:
All parts of this paper seem to be too long and redundant. It can be written much more concisely. Moreover, a thorough review by a native English speaker would be helpful.
A thorough review by a native English speaker would be helpful.
Author Response
Thank you very much for the kind evaluation and all the valuable comments. We made our every effort to address all the concerns one-by-one by the Reviewers, and we have deleted some redundant content to refine the manuscript.
We apologize for the language problem, we have polished the language using a language editing service(https://webshop.elsevier.com/language-editing-services/language-editing/) to improve the logic and readability before our original submission. And, the revised manuscript was proofread and edited by a native English speaker. We hope the reviewer will be satisfied with the revised manuscript.
Round 2
Reviewer 1 Report
The authors addressed all of the comments. Below is the list of suggestions for the final, minor edits:
1. Line 39: format “2” in O2•– as a subscript
2. Lines 40-42: “…. do not possess unpaired electron but still… molecular oxygen [5-6].” - consider rewording to “… do not possess unpaired electron but are the downstream products of incomplete reduction of molecular oxygen and have oxidizing potential [5-6].”
3. Line 47 – include reduced glutathione in the list of non-enzymatic antioxidants.
4. Lines 101, 113, 228, 241 – correct a double dot at the end of sentence.
5. Lines 121, 122, 176, 178: “2+” in Fe2+ should be a superscript.
6. Line 184 – “H2O2” – use subscripts for the numbers
7. Line 342 still refers to Supplementary Figure 1. In their response to previous review the authors indicated they moved the figure to the main text.
8. Lines 397-399 – two sentences describe the same drug, but the beginning of the second sentence suggest a different/another drug is described.
9. Line 437: change “by ROS levels” to “from ROS”
Author Response
Thank you very much for the kind evaluation and all the valuable comments. We provide a point-by-point response below. And we have revised the manuscript carefully following the comments. Thanks again.
- Line 39: format “2” in O2•– as a subscript
Response 1: Thanks for the kind reminder. Corresponding modification has been made in Page 2, line 40 in the revised manuscript.
- Lines 40-42: “…. do not possess unpaired electron but still… molecular oxygen [5-6].” - consider rewording to “… do not possess unpaired electron but are the downstream products of incomplete reduction of molecular oxygen and have oxidizing potential [5-6].”
Response 2: Thank you for the valuable comment. Corresponding modification has been made in Page 2, line 40 to line 42.
- Line 47 – include reduced glutathione in the list of non-enzymatic antioxidants.
Response 3: Thank you for the suggestion. We have made corresponding modification in Page 2, line 49.
- Lines 101, 113, 228, 241 – correct a double dot at the end of sentence.
Response 4: Thank you for your patience and careful reading. We have rechecked the manuscript and deleted the unnecessary dots.
- Lines 121, 122, 176, 178: “2+” in Fe2+ should be a superscript.
Response 5: Thanks for the kind reminder. Corresponding modification has been made in Page , line 127 128 186 189 in the revised manuscript.
- Line 184 – “H2O2” – use subscripts for the numbers
Response 6: Thanks for the kind reminder. Corresponding modification has been made in Page , line 195 in the revised manuscript.
- Line 342 still refers to Supplementary Figure 1. In their response to previous review the authors indicated they moved the figure to the main text.
Response: Thank you very much for the kind remainder. We sincerely apologize for this omission. We have moved the previous Supplementary Figure 1 to the main text as Figure 3 in Page 14. Corresponding modification could be found in the revised manuscript(Page 11, line 362).
- Lines 397-399 – two sentences describe the same drug, but the beginning of the second sentence suggest a different/another drug is described.
Response: Thank you for the helpful comment. We apologize for the confusion. Imidazole-based compounds are a subset of azole-based derivatives, the latter encompasses a broader range of compounds that include other azole derivatives as well. To avoid any confusion, we rephrased the statement as follow (Page 12, line 408 to line 410): “Azole-based derivatives such as ketoconazole, terconazole, sulconazole nitrate, and imidazole-based compounds used for HO-1 inhibitors are also reported”.
- Line 437: change “by ROS levels” to “from ROS”
Response: Thank you very much for the helpful comment. Corresponding modification has been made in Page 14 line 469.
Reviewer 3 Report
The manuscript was further improved and is acceptable for publication in its present form
Author Response
Thank you for recommending our manuscript for publication. We greatly appreciate your insightful and valuable comments in helping us improve the manuscript.
Reviewer 4 Report
I have no further comments.
Author Response
Thank you very much for your insightful and valuable comments in helping us improve the manuscript.